# Associations of Warfarin Use with Risks of Ischemic Cerebrovascular Events and Major Bleeding in Patients with Hyperthyroidism-Related Atrial Fibrillation

**DOI:** 10.3390/biomedicines10112670

**Published:** 2022-10-22

**Authors:** Sian-De Liu, Shwu-Jiuan Lin, Chin-Ying Ray, Fang-Tsyr Lin, Weei-Chin Lin, Li-Hsuan Wang

**Affiliations:** 1School of Pharmacy, Taipei Medical University, Taipei 110, Taiwan; 2Department of Pharmacy, New Taipei Municipal TuCheng Hospital (Built and Operated by Chang Gung Medical Foundation), New Taipei City 236, Taiwan; 3PhD Program in Clinical Drug Development of Herbal Medicine, College of Pharmacy, Taipei Medical University, Taipei 110, Taiwan; 4Department of Clinical Pharmacy, Chang Gung Memorial Hospital, Linkou, Taoyuan 333, Taiwan; 5Heart Failure Center, Chang Gung Memorial Hospital, Linkou, Taoyuan 333, Taiwan; 6Section of Hematology/Oncology, Department of Medicine, Baylor College of Medicine, Houston, TX 77030, USA; 7Department of Molecular and Cellular Biology, Baylor College of Medicine, Houston, TX 77030, USA; 8Department of Pharmacy, Taipei Medical University Hospital, Taipei 110, Taiwan

**Keywords:** atrial fibrillation, hyperthyroidism, warfarin, ischemic stroke, transient ischemic attack

## Abstract

The use of oral anticoagulants for patients with new-onset hyperthyroidism-related atrial fibrillation (AF) is controversial. We aimed to evaluate the clinical benefits of warfarin therapy in this population. This retrospective cohort study used a data-cut of Taiwan Health and Welfare Database between 2000 and 2016. We compared warfarin users and nonusers among AF patients with hyperthyroidism. We used 1:2 propensity score matching to balance covariates and Cox regression model to calculate hazard ratios (HRs). The primary outcome was risk of ischemic stroke/transient ischemic attack (TIA), and the secondary outcome was major bleeding. After propensity score matching, we defined 90 and 168 hyperthyroidism-related AF patients with mean (SD) age of 59.9 ± 13.5 and 59.2 ± 14.6 in the warfarin-treated group and untreated group separately. The mean (SD) CHA2DS2-VASc scores for the two groups were 2.1 ± 1.6 and 1.8 ± 1.5, respectively. Patients with hyperthyroidism-related AF receiving warfarin had no significant risk of ischemic stroke/TIA (adjusted HR: 1.16, 95% confidence interval [CI]: 0.52–2.56, *p* = 0.717) compared to nonusers. There was a comparable risk of major bleeding between those receiving warfarin or not (adjusted HR: 0.91, 95% CI: 0.56–1.47, *p* = 0.702). The active-comparator design also demonstrated that warfarin use had no significant association with the risk of stroke/TIA versus aspirin use (adjusted HR: 2.43; 95% CI: 0.68–8.70). In conclusion, anticoagulation therapy did not have a statistically significant benefit on ischemic stroke/TIA nor risk of bleeding, among patients with new-onset hyperthyroidism-related AF under a low CHA2DS2-VASc score, by comparing those without use.

## 1. Introduction

Atrial fibrillation (AF) is a common sustained cardiac arrhythmia in adults worldwide, and it is associated with substantial mortality and risk of ischemic stroke [1]. AF contributes to an overall five-fold higher risk of stroke [2]. It is well recognized that hyperthyroidism is associated with a risk of heart disease progression, but by itself can cause cardiac disease [3,4]. Hyperthyroidism is consistently reported as a risk factor for AF [5,6]. Thyroid hormones modulate the transcription rate of multiple genes, the production of sarcoplasmic reticulum proteins, calcium-activated ATPase and phospholamban in cardiac myocytes [3,7]. Furthermore, the hormones increase in systolic depolarization and diastolic repolarization, decreasing the action potential duration and the refraction period of the atrial myocardium, as well as the atrial nodal refraction period [7]. Elevated thyroid hormones activate arrhythmogenic foci and increase supraventricular ectopic activity, which is considered a significant causal link between hyperthyroidism and AF [7,8,9]. Additionally, to identify those at high risk for incident AF, hyperthyroidism was proposed as a variable involved in the C2HEST score (coronary artery disease or chronic obstructive pulmonary disease (1 point each), hypertension (1 point), elderly (aged ≥ 75 years, 2 points), systolic heart failure (2 points), thyroid disease (hyperthyroidism, 1 point)) [2,6,10]. However, its correlation with thromboembolic stroke among patients with hyperthyroidism-related AF is still unclear [3,11,12,13,14]. Prophylactic treatment with oral anticoagulants for stroke prevention in this population is still controversial.

A previous prospective study in hyperthyroid patients with new-onset AF reported an increased risk of ischemic stroke clustering during the presentation phase [11]. Those authors suggested prompt early use of anticoagulation therapy in such patients with hyperthyroidism-related AF [11]. Another study showed that warfarin was beneficial compared to aspirin or no-treatment in stroke prevention in these patients with a CHA2DS2-VASc score of ≥1 and persistent AF [12]. However, the presence of hyperthyroidism did not confer an additional risk of ischemic stroke compared to that with non-hyperthyroid AF [12]. Recent research suggested that thromboprophylaxis with direct oral anticoagulants (DOACs) may be an effective and safer alternative to warfarin and should be considered for patients with AF concomitant with hyperthyroidism [15]. 

Current guidelines present limited evidence on management of AF in these settings [1,2,16]. Some studies revealed that hyperthyroidism is not an independent higher risk factor for stroke or systemic embolic events compared to non-hyperthyroid patients [12,13,14]. There are currently no recommendations focusing on AF patients with hyperthyroidism. However, Canadian guidelines suggest anticoagulation therapy during thyrotoxicosis with low-quality evidence [17]. To the present, no randomized controlled trial has specifically focused on these patients in relation to the efficacy and safety of anticoagulation therapy. Given the lack of clear evidence, a recommendation to initiate anticoagulation appears to be warranted. Therefore, the aim of our study was to investigate the association between warfarin therapy and risks of ischemic stroke/transient ischemic attack (TIA) and bleeding among patients with hyperthyroidism-related AF.

## 2. Materials and Methods

### 2.1. Source of Data

This retrospective cohort study used a subset of the Health and Welfare Database (HWD) from Health and Welfare Data Science Center of Taiwan. The HWD consists of medical claim data of the National Health Insurance which covers 99.99% of Taiwan’s population. We used data from the 2000 Longitudinal Generation Tracking Database (LGTD 2000), which contains the information of 2 million beneficiaries randomly sampled from the HWD. Details of the HWD and the subset were documented in previous studies [18,19,20,21]. The subset contains all anonymized and deidentified personal information, medical records, procedures, and diagnoses, which were recorded using the International Classification of Diseases, 9th Revision, Clinical Modification (ICD-9-CM) codes from 1997 through 2015 and the International Classification of Diseases, 10th Revision, Clinical Modification (ICD-10-CM) codes since 2016. This study obtained ethical approval from the Taipei Medical University Joint Institutional Review Board (no. N201908068; 31 August 2019) and followed the Strengthening the Reporting of Observational Studies in Epidemiology (STROBE) guidelines.

### 2.2. Study Population

We extracted data on patients who aged ≥20 years and had one discharge nonvalvular AF diagnosis or two or more records in an outpatient department [18,20]. Patients with hyperthyroidism were identified by having two consecutive records of a diagnosis [22]. Individuals were all from January 2002 through December 2012 with at least a 2-year washout period to define newly diagnosed cases. Among these patients, those whose first nonvalvular AF diagnosed with hyperthyroidism was within 1 year were defined as hyperthyroidism-related nonvalvular AF.

Exclusion criteria for the study were patients (1) who had a stroke or TIA; (2) who had previous major bleeding; or (3) who had a diagnosis of hypothyroidism before the date of the hyperthyroidism diagnosis to maximize specificity for a diagnosis of hyperthyroidism. Details of the study design flowchart and patient numbers are given in Figure 1.

### 2.3. Medication Exposure and Covariates

We defined patients with hyperthyroidism-related nonvalvular AF who received warfarin within 365 days after the first nonvalvular AF diagnosis as the exposed group. The index date was defined as the first date on which warfarin was prescribed. A total prescription length of 7 days or longer was applied. We excluded patients who had previously used anticoagulants. All enrolled patients were followed up from the index date until the occurrence of any study outcome independently, or until the end date of the study (31 December 2016), whichever came first.

Patient demographics, comorbidities (hypertension, myocardial infarction, congestive heart failure, peripheral vascular disease, cerebrovascular disease, diabetes mellitus, chronic obstructive pulmonary disease, rheumatic disease, chronic kidney disease, hyperlipidemia, cardiomyopathy, pulmonary embolism, deep vein thrombosis, atherosclerosis, and any malignancy including leukemia) were identified as covariates. The CHA2DS2-VASc score was calculated as a measure of stroke risk. Relevant medications (amiodarone, aspirin, P2Y12 inhibitors, nonsteroidal anti-inflammatory drugs, and statins) were also assessed. The code lists of these covariates are shown in Appendix A.

### 2.4. Outcomes

The primary outcome was a composite of ischemic stroke and TIA. The diagnostic accuracy of ischemic stroke in the HWD is high [23,24,25]. We designated events first occurring after 14 days following the index date as outcomes, because the occurrence of events within the first few days of a diagnosis of AF was most likely related to the initial presentation of AF rather than a new event. The secondary outcome was major bleeding, defined as intracranial hemorrhage, or bleeding at gastrointestinal or other sites requiring hospitalization. We included only one major bleeding event occurring after the index date [20]. The code lists of these outcomes are shown in Appendix A.

### 2.5. Patient and Public Involvement

Patients or the public were not involved in the design, or conduct, or reporting or dissemination plans of our research.

### 2.6. Statistical Analysis

Continuous variables are expressed as the mean (±standard deviation (SD)), and categorical variables are expressed as proportions. Differences between continuous values were compared using a two-tailed *t*-test, and differences between nominal variables were compared using a Chi-squared test. We used a propensity score (PS) matching method to balance covariates across the warfarin-treated and untreated groups [26]. The PS was estimated using a multivariable logistic regression model based on various patient baseline characteristics which are listed in Table 1. We applied the greedy nearest neighbor 1:2 PS matching with a caliper of 0.2 on the PS scale, and an exact match was made for the index year [26]. The balance of baseline characteristics was evaluated using the standardized mean difference (SMD). Characteristics with an absolute SMD of >0.2 following PS matching were considered imbalanced between the two groups [27]. After PS matching, Cox proportional hazard modeling with a robust sandwich variance estimator analysis was performed to compare rates of clinical events [28]. 

We performed the first sensitivity analysis to examine the impacts of DOACs and new antiplatelet medication users on outcomes, because the approval date for the first DOACs in Taiwan was June 2012. Second, to reduce potential unmeasured confounding, we performed an active comparator design with warfarin versus aspirin. All statistical analyses were conducted by using SAS 9.4 (version 9.4, SAS Institute, Cary, NC, USA). Statistical significance was defined as a two-tailed *p* value of < 0.05.

## 3. Results

### 3.1. Baseline Characteristics

From a total of 447 eligible patients included in the study, 105 (23.5%) were new warfarin users, and 342 individuals (76.5%) were nonusers. Table 1 shows characteristics of the unmatched and matched cohorts. Before PS matching, warfarin users had significantly higher CHA2DS2-VASc scores than did the untreated group. In addition, warfarin users were significantly more likely to have a history of comorbidities (diabetes mellitus, heart failure, myocardial infarction). After PS matching, 258 patients were included in the final analysis, with 90 patients in the treated group (mean (SD) age, 59.9 (13.5) years; 38 males (42.2%)) and 168 patients in the untreated group (mean (SD) age, 59.2 (14.6) years; 75 males (44.6%)). The mean (SD) CHA2DS2-VASc scores were 2.1 (1.6) and 1.8 (1.5) in the warfarin-treated and untreated groups, respectively. Most patient characteristics were comparable between the two groups after PS matching (SMD < 0.2), except the history of hypertension and heart failure, which were further added in the regression model.

### 3.2. Risks of an Ischemic Stroke and Transient Ischemic Attack

The primary study outcome is summarized in Table 2. In the matched cohort, there were 19 events (21.1%) in the warfarin-treated group and 21 events (12.5%) in the untreated group. Mean follow-up times of stroke/TIA occurrence were 6.7 years among warfarin users and 7.2 years among nonusers. In terms of stroke/TIA events, the annual cumulative incidence was 1.74% per year in hyperthyroidism-related AF among nonusers. Results from the robust Cox regression model with multiple covariates are presented in Appendix A. The risk of stroke/TIA among warfarin users did not significantly differ from that of nonusers (adjusted hazard ratio (aHR): 1.16, 95% confidence interval (CI): 0.52–2.56). Compared to the warfarin-untreated group, congestive heart failure (aHR: 4.30, 95% CI: 1.15–16.09), peripheral vascular disease (aHR: 6.01, 95% CI: 1.56–23.11) and a history of taking aspirin (aHR: 3.42, 95% CI: 1.37–8.54) were significantly associated with an increased risk of stroke/TIA.

### 3.3. Risk of Major Bleeding

The secondary study outcome is summarized in Table 2. In the matched cohort, there were 39 events (43.3%) in the warfarin-treated group and 64 events (38.1%) in the untreated group. Mean follow-up times of major bleeding occurrence among both groups were about 5.4 years. Results of the robust Cox regression model with multiple covariates are presented in Appendix A. The risk of major bleeding among warfarin users did not significantly differ from that of nonusers (aHR: 0.91, 95% CI: 0.56–1.47). Compared to the warfarin-untreated group, hypertension (aHR: 2.11, 95% CI: 1.15–3.88) and a history of taking aspirin (aHR: 1.85, 95% CI: 1.20–2.85) were significantly associated with an increased risk of major bleeding.

### 3.4. Sensitivity Analyses

The results of the first sensitivity analysis revealed consistency in the findings even when we excluded DOACs and new antiplatelet medication users after the index date. After multivariate adjustment, the risk of the occurrence of stroke/TIA did not significantly differ between the warfarin users and nonusers (Appendix A, aHR, 1.18, 95% CI: 0.53–2.64). The risk of the occurrence of bleeding had no significant association in the warfarin users (aHR: 0.95, 95% CI: 0.58–1.55). To reduce selection bias, we repeated our analysis with use of warfarin versus aspirin (Appendix A). The results are in accordance with our earlier observations. It demonstrated that warfarin use had no significant association with the risk of stroke/TIA, even comparing with aspirin use (aHR: 2.43, 95% CI: 0.68–8.70).

## 4. Discussion

This is the first retrospective cohort study that compares risks of ischemic stroke/TIA and major bleeding associated with warfarin users vs. nonusers among patients with hyperthyroidism-related AF. In the clinical reality, new onset disease might not require anticoagulation therapy. Our results showed that the use of warfarin was not associated with a statistically significantly lower risk of ischemic stroke/TIA nor a higher risk of major bleeding. The sensitivity analysis which excluded new DOAC users showed similar results.

In our study, 2.8% of patients referred to the hospital with AF also had hyperthyroidism. This rate is compatible with other studies in Taiwan and other countries [14,29,30], and female patients were predominant [12,15,30]. The annual stroke/TIA incidence of 1.74% per year among nonusers with a mean CHA2DS2-VASc score of 1.8 was numerically lower in Taiwanese and in a Swedish cohort which had a CHA2DS2-VASc score of 2 [13,31]. The reason could be that patients with hyperthyroidism-related AF are not at a higher risk of ischemic stroke than those with non-thyroid AF [12,30]. In most AF patients with hyperthyroidism, spontaneous reversion to sinus rhythm often occurs after restoration of thyroid function [2,29]. Antithyroid therapy were prescribed to decrease the arrhythmogenic activity under the guidance of Taiwan National Health Insurance program [9,16,30]. However, a national cohort study demonstrated the incidence of thromboembolic events may be not influenced by antithyroid therapies among patients with hyperthyroidism-related AF [30]. Another study used a Cox proportional hazard model to predict that over 60% of patients who cardioverted even after 4 to 10 years of AF will remain in sinus rhythm when so treated [29]. Consistent with our data, the duration of stroke/TIA was compatible. This result probably implies that hyperthyroidism in new onset AF patients with a CHA2DS2-VASc score of <2 would have less influence on short-term outcomes.

Most studies showed that hyperthyroidism per se did not confer an additional risk of thromboembolic events among patients with AF compared to non-hyperthyroidism patients [14,30,32]. Our study was conducted to determine the role of anticoagulation therapy in patients with hyperthyroidism-related AF through a mimic clinical controlled trial. The finding of no association between warfarin use (users vs. nonusers) and ischemic stroke/TIA is also consistent with several previous observational studies conducted with other study designs and in other populations [12,14,30,32]. An early study with age-matched controls found that the risk of cerebrovascular events in thyrotoxicosis patients with AF did not significantly increase compared to those in sinus rhythm [14]. However, the mean ages of the two groups were markedly different at 56 and 39 years, respectively. Moreover, a recent single-center observational study demonstrated that in hyperthyroidism patients with non-paroxysmal AF, warfarin therapy was associated with a reduced risk of ischemic stroke via Kaplan-Meier estimate. However, no statistical association was found in univariate analysis [12]. Nevertheless, those studies did not adjust for amiodarone use, [12,14,32], and those positive results were based on the patients with old age and a vary-high risk of thromboembolism (high CHA2DS2-VASc score) [12,15,33,34].

Conversely, our population was less severe in terms of both new onset of AF and hyperthyroidism with mean CHA2DS2-VASc scores of around 2, which is close to clinical practice, when the first diagnosis and a decision as to whether to initiate anticoagulation therapy are made. The two groups in our study had comparable ages, genders, and mean CHA2DS2-VASc scores, and other different baseline comorbidities were included in the Cox model for adjustment. In addition, a cohort study showed that DOAC use was associated with a comparable risk of thromboembolisms and a significantly lower risk of major bleeding compared to warfarin use [15]. So, we excluded DOAC use in the two groups, and thus there was no association with outcome risks and warfarin use compared to nonusers.

The results of the study have clinical implications. We demonstrated that anticoagulation therapy did not have a statistically significant benefit among patients with new-onset AF that occurred with new-onset hyperthyroidism compared to those with no anticoagulation therapy. Annual cerebral thromboembolic event rates in the untreated group were lower than those in Taiwanese and Swedish cohorts [13,31]. Therefore, oral anticoagulation might not be required in terms of cerebral thromboprophylaxis, especially in patients with a first episode and who are younger than 65 years. Our data support the gap of current guidelines [2,35], and the initiation of oral anticoagulants was based on the CHA2DS2-VASc score despite hyperthyroidism.

Our study has some limitations. First, this is not a prospective randomized study. Second, the cohort study was based on ICD codes that could be limited by coding errors. However, diagnoses of AF, hyperthyroidism, stroke, bleeding, and other comorbidities in the HWD are well validated [18,20,22,23,24,25]. Third, information on different types of AF is not available in the HWD. Forth, we had no data of antithyroid medications and the exact timing of recovery to sinus rhythms. However, a national retrospective cohort study found that different antithyroid medications did not influence the incidence of thromboembolic events in this population [30]. This supports our study results that antithyroid medications had little impact on our outcomes. Fifth, we did not apply an active comparator design and might have experienced confounding by indications (high stroke/TIA rates). Nevertheless, there is no other appropriate active drug used in clinical practice in terms of oral anticoagulants, and most patients realistically receive no anticoagulants in Taiwan [30]. Although some selection bias might remain, we adjusted for confounding biases as far as was possible, even accounting for different baseline comorbidities between the two groups. We used PS matching and tried to adjust for baseline differences using variant multivariable Cox proportional hazard regression analyses to minimize the bias. Sixth, data for different types of AF are not available in the HWD. However, hyperthyroidism-related AF has a higher chance of paroxysmal AF, and previous studies have indicated that patients with paroxysmal AF might suffer fewer thromboembolic events when compared with patients with non-paroxysmal AF [36]. Seventh, HWD did not have laboratory data on international normalized ratios (INRs). However, up to 99.99% of Taiwan’s population was enrolled under National Health Insurance Program. Patients had good access to health care, and their physicians prescribed warfarin based on the clinical presentation and INR monitoring. Therefore, we assumed that the individual differences in the INR were minimal. Finally, the actual drug consumption could not be determined from the database. Therefore, our conclusion was based on reasonable therapeutic adherence.

## 5. Conclusions

Among patients with hyperthyroidism-related AF, oral anticoagulation therapy was not associated with benefits in terms of ischemic cerebrovascular events or with major bleeding. The findings suggest that patients who are younger than 65 years with new-onset hyperthyroidism-related AF do not require an oral anticoagulant under a CHA2DS2-VASc score < 2. The initiation of oral anticoagulants should be based on CHA2DS2-VASc scores according to current guidelines. Healthcare providers should consider periodic reassessment of the AF and thyroid statuses. Future large controlled studies of oral anticoagulants for preventing stroke in such patients are required to confirm these findings.

## Figures and Tables

**Figure 1 biomedicines-10-02670-f001:**
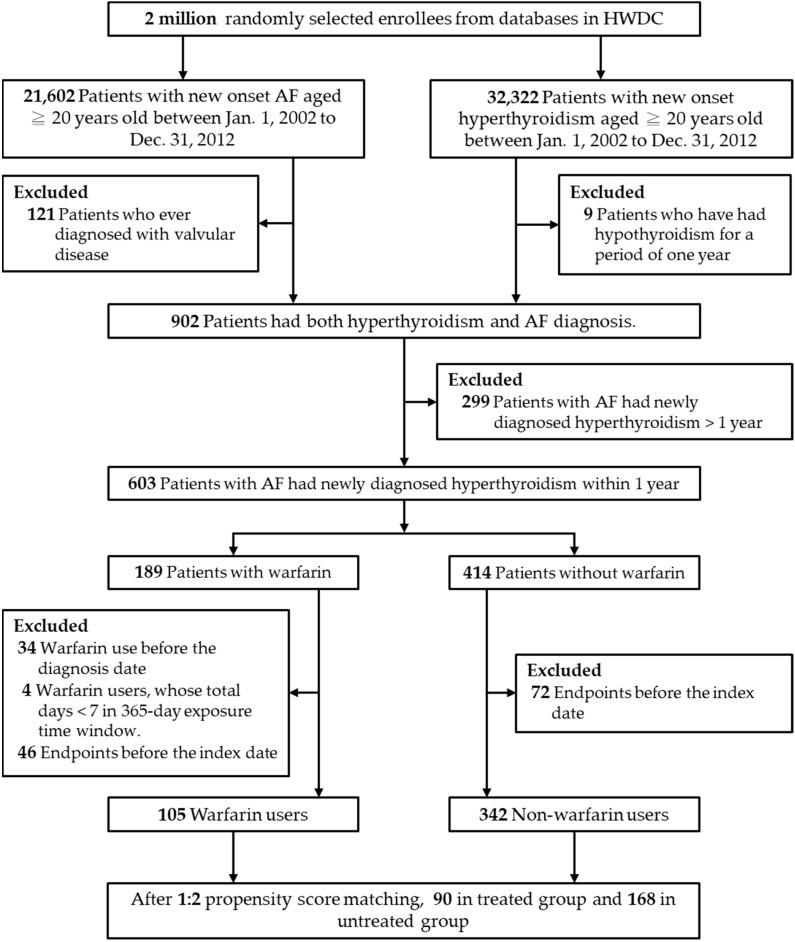
Flowchart of the study and analysis. Abbreviations: AF, atrial fibrillation; HWDC, Health and Welfare Data Science Center.

**Table 1 biomedicines-10-02670-t001:** Baseline characteristics of the study population.

Characteristic (*n*, %)	Before PSM				After PSM			
	Warfarin Users	Warfarin Nonusers	*p* Value	SMD	Warfarin Users	Warfarin Nonusers	*p* Value	SMD
	*N* = 105	*N* = 342			*N* = 90	*N* = 168		
Age, years (mean ± SD)	60.1 ± 12.9	57.4 ± 14.5	0.088	0.197	59.9 ± 13.5	59.2 ± 14.6	0.699	0.051
Aged ≥ 65 years	40 (38.1)	111 (32.5)	0.285	0.118	36 (40.0)	64 (38.1)	0.765	0.039
Gender: Male	48 (45.7)	124 (36.3)	0.082	0.118	38 (42.2)	75 (44.6)	0.709	0.039
CHA2DS2-VASc (mean ± SD)	2.0 ± 1.6	1.7 ± 1.4	0.080	0.205	2.1 ± 1.6	1.8 ± 1.5	0.171	0.178
Hypertension	38 (36.2)	93 (27.2)	0.076	0.194	32 (35.6)	43 (25.6)	0.093	0.218
Diabetes mellitus	22 (21.0)	44 (12.9)	0.041	0.217	18 (20.0)	27 (16.1)	0.428	0.102
Hyperlipidemia	9 (8.6)	36 (10.5)	0.560	−0.067	9 (10.0)	20 (11.9)	0.644	−0.061
Chronic kidney disease	3 (2.9)	17 (5.0)	0.433	−0.109	2 (2.2)	4 (2.4)	0.936	−0.011
Congestive heart failure	38 (36.2)	58 (17.0)	<0.0001	0.446	35 (38.9)	44 (26.2)	0.035	0.274
Myocardial infarction	0	10 (2.9)	0.126	−0.245	0	0	NA	NA
Peripheral vascular disease	4 (3.8)	4 (1.2)	0.092	0.170	3 (3.3)	4 (2.4)	0.698	0.057
Chronic obstructive pulmonary disease	14 (13.3)	37 (10.8)	0.478	0.077	13 (14.4)	21 (12.5)	0.660	0.057
Rheumatic disease	0	3 (0.9)	1.000	−0.133	0	2 (1.2)	0.544	−0.155
Thromboembolism	0	1 (0.3)	1.000	−0.077	0	0	NA	NA
Cardiomyopathy	4 (3.8)	5 (1.5)	0.224	0.147	4 (4.4)	3 (1.8)	0.243	0.154
Any malignancy including leukemia	0	5 (1.5)	0.596	−0.172	0	3 (1.8)	0.554	−0.191
Cerebrovascular disease	2 (1.9)	4 (1.2)	0.629	0.060	2 (2.2)	3 (1.8)	1.000	0.031
Amiodarone	20 (19.1)	50 (14.6)	0.275	0.119	19 (21.1)	24 (14.3)	0.161	0.180
Aspirin	38 (36.2)	105 (30.7)	0.292	0.117	35 (38.9)	50 (29.8)	0.137	0.193
Clopidogrel	3 (2.9)	9 (2.6)	1.000	0.014	3 (3.3)	2 (1.2)	0.346	0.145
NSAIDs	10 (9.5)	34 (9.9)	0.900	−0.014	9 (10.0)	18 (10.7)	0.858	−0.023
Statins	6 (5.7)	16 (4.7)	0.668	0.047	6 (6.7)	9 (5.4)	0.668	0.055

Abbreviations: NA, not applicable; NSAIDs, non-steroid anti-inflammatory drugs; PSM, propensity score matching; SD, standard deviation; SMD, standardized mean difference.

**Table 2 biomedicines-10-02670-t002:** Number of events and adjusted hazard ratios between warfarin users and nonusers among patients with hyperthyroidism-related atrial fibrillation.

	After PSM		
	Warfarin Users	Warfarin Nonusers	*p* Value
	*N* = 90	*N* = 168	
**Stroke/TIA**			
Events (%)	19 (21.1)	21 (12.5)	0.069
Mean follow-up time (months, SD)	79.8 (43.1)	86.0 (41.1)	0.254
aHR (95% CI) ^a^	1.16 (0.52–2.56)	Reference	0.717
**Major bleeding**			
Events (%)	39 (43.3)	64 (38.1)	0.413
Mean follow-up time (months, SD)	65.0 (45.5)	65.3 (43.7)	0.958
aHR (95% CI) ^b^	0.91 (0.56–1.47)	Reference	0.702

Abbreviations: CI, confidence interval; PSM, propensity score matching; SD, standard deviation; TIA, transient ischemic attack. ^a^ Adjusted for age, gender, CHA2DS2-VASc, hypertension, congestive heart failure, peripheral vascular disease, diabetes mellitus, and aspirin. ^b^ Adjusted for age, gender, CHA2DS2-VASc, hypertension, congestive heart failure, diabetes mellitus, chronic obstructive pulmonary disease, hyperlipidemia, aspirin, and statins.

## Data Availability

We must follow the Computer-Processed Personal Data Protection Law and related regulations in Taiwan. The datasets for this manuscript are not publicly available because of the HWD protection policy.

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
