# Peer review of "Associations of Warfarin Use with Risks of Ischemic Cerebrovascular Events and Major Bleeding in Patients with Hyperthyroidism-Related Atrial Fibrillation"

_biomedicines, 2022, doi:10.3390/biomedicines10112670_

Round 1

Reviewer 1 Report

Liu et al are presenting an interesting study aiming to evaluate the benefits of warfarin in patients with hyperthyroidism-related atrial fibrillation. The scientific interest is unquestionable and the manuscript very easy to read. The authors acknowledge the study main limitations, particularly the sample size. I do have the following comments:

Major

1.“This implies that anticoagulation therapy might not be necessary. Conversely, Cana[1]dian guidelines suggest anticoagulation therapy during thyrotoxicosis with low-quality evidence [14].”  This sentence is not accurate. Firstly, there is no underlying message or implicit recommendation against anticoagulation in patients with AF and hyperthyroidism.

2. Therapeutic adherence was not evaluated. The authors should discuss this limitation.

3. Treatment of hyperthyroidism might decrease the associated arrhythmogenic, including remission of AF in younger patients. The authors should discuss this particular aspect, and develop the discussion around the risk associated with paroxysmal vs permanent AF.

Minor

1.“ the present, no randomized controlled trial has specifically focused on these patients in relation to the efficacy and safety of anticoagulation therapy. “ – It is really very hard to understand why the authors would want an RCT specially focused on the efficacy of anticoagulation in patients with AF and hyperthyroidism. The introduction should be more focused on the mechanisms of AF and atrial cardiopathy in hyperthyroidism.

2. Detail how users (anticoagulation group) were defined – short term use ? regular use ?

Reviewer 2 Report

This study used the Health and Welfare Database (HWD) from Health and Welfare Data Science Center of Taiwan to retrospectively select the potential subjects. The detailed inclusion and exclusion criteria for the subjects were provided. From 2002 to 2012, 447 patients had both hyperthyroidism and AF diagnosis, where 105 patients were warfarin users. To have a compatible population, the subjects between two groups were matched by the 1:2 propensity score. At the end, 90 subjects treated with warfarin and 168 subjects were untreated. The participants were correctly collected. However, the major flaw of this study was the analyses for the survival part. To be specific, the event of interest was a composite of ischemic stroke and TIA. The outcome of interest was the duration from the index date to the event. The authors used the data from 2002 to 2012 and computed the duration till the occurrence of the study outcome or the end date of the study (12/31/2016). The participant enrolled in 2002 would be followed for almost 15 years, whereas those enrolled in 2012 would be only followed for less than 5 years. The analyses would be only valid for the survival time less than 5 years. However, the mean follow-up time was 79.8 and 86 months for warfarin users and nonusers. Since the event rate is relatively low, it would be a bit difficult to understand the value of this study. Also, some of the controlling covariates in the logistic model for building the propensity scores have very low frequencies which would cause some computational problems when finding the estimates of the coefficients. These variables should not be put in the models. 

Reviewer 3 Report

The authors focused on the study Associations of warfarin use with risks of ischemic cerebrovascular events and major bleeding in patients with hyperthyroidism-related atrial fibrillation. This is an interesting study, but there are some important facts that the authors should add. The tables and figure in the text are very clearly written.

In my opinion:

- The abstract presents an inaccurate description of this study.

- The author has performed an inadequate literature review (recent reports from the field are missing).

- The patients are described adequately.

- The management of the study is not effectively described (doses of warfarin are missing, what about aspirin? how many times a day the drugs were administered). After filling in the missing data, I propose to complete the discussion.

Round 2

Reviewer 1 Report

The authors have responded to all my comments. The manuscript is now suitable for publication.

Reviewer 3 Report

The authors have responded to all my comments.

In my opinion the manuscript is now suitable for publication.